# Imaging magnetic transition of magnetite to megabar pressures using quantum sensors in diamond anvil cell

Mengqi Wang [1,2,3,9], Yu Wang [4,5,9] ✉, Zhixian Liu[1,2,3,9], Ganyu Xu[1,2,3], Bo Yang[1,2,3], Pei Yu [1,2,3], Haoyu Sun[1,2,3], Xiangyu Ye [1,2,3], Jingwei Zhou [1,2,3,6], Alexander F. Goncharov [7], Ya Wang [1,2,3,6] ✉ & Jiangfeng Du [1,2,3,6,8] ✉

High-pressure diamond anvil cells have been widely used to create novel states of matter. Nevertheless, the lack of universal in-situ magnetic measurement techniques at megabar pressures makes it difficult to understand the underlying physics of materials' behavior at extreme conditions, such as high-temperature superconductivity of hydrides and the formation or destruction of the local magnetic moments in magnetic systems. Here, we break through the limitations of pressure on quantum sensors by modulating the uniaxial stress along the nitrogen-vacancy axis and develop the in-situ magnetic detection technique at megabar pressures with high sensitivity ($\sim 1\mu T/\sqrt{Hz}$) and sub-microscale spatial resolution. By directly imaging the magnetic field and the evolution of magnetic domains, we observe the macroscopic magnetic transition of $Fe_3O_4$ in the megabar pressure range from ferrimagnetic ($\alpha$-$Fe_3O_4$) to weak ferromagnetic ($\beta$-$Fe_3O_4$) and finally to paramagnetic ($\gamma$-$Fe_3O_4$). The scenarios for magnetic changes in $Fe_3O_4$ characterized here shed light on the direct magnetic microstructure observation in bulk materials at high pressure and contribute to understanding magnetism evolution in the presence of numerous complex factors such as spin crossover, altered magnetic interactions and structural phase transitions.

Pressure has been proved to be a powerful tool to tune the magnetic properties of materials as it can effectively increase intermolecular interaction and redistribute electrons[1-3]. For example, the spin crossover in 3d transition metal compounds is characterized by the competition of the spin exchange and crystal field energy on compression[4], and the effect of covalency of the neighboring metal ions[5].

Additionally, measurements of magnetic properties at high pressures are extremely instrumental in high $T_c$ superconducting materials such as polyhydrides, where there is a high controversy about their existence and properties[6-8] due to a lack of sufficient evidence of the Meissner effect. Many efforts have been made to adapt magnetic detection methods to Diamond Anvil Cell (DAC) facilities. High energy

[1]CAS Key Laboratory of Microscale Magnetic Resonance and School of Physical Sciences, University of Science and Technology of China, Hefei 230026, China. [2]Anhui Province Key Laboratory of Scientific Instrument Development and Application, University of Science and Technology of China, Hefei 230026, China. [3]CAS Center for Excellence in Quantum Information and Quantum Physics, University of Science and Technology of China, Hefei 230026, China. [4]Key Laboratory of Materials Physics, Institute of Solid State Physics, HFIPS, Chinese Academy of Sciences, Hefei, China. [5]Institute of Geosciences, Goethe University Frankfurt, Frankfurt 60438, Germany. [6]Hefei National Laboratory, University of Science and Technology of China, Hefei 230088, China. [7]Earth and Planets Laboratory, Carnegie Institution of Washington, Washington, DC, USA. [8]Institute of Quantum Sensing and School of Physics, Zhejiang University, Hangzhou 310027, China. [9]These authors contributed equally: Mengqi Wang, Yu Wang, Zhixian Liu. ✉e-mail: wangyu@issp.ac.cn; ywustc@ustc.edu.cn; djf@ustc.edu.cn

photon/neutron scattering techniques such as x-ray magnetic circular dichroism (XMCD), Neutron magnetic scattering, and Mössbauer spectroscopy are widely used for magnetic measurements in DACs[1]. These techniques can probe atomic-scale magnetism and resolve local element-specific magnetism issues. However, these methods are not directly sensitive to magnetic phenomena like the Meissner effect. Moreover, they are technically restricted by sample and beam size at high pressure, and the quantitative analysis remains challenging[9]. The magnetization of the materials can be directly measured by integrating DAC into superconducting quantum interference devices (SQUID). However, the effect of the DAC apparatus' background cannot be separated from the magnetization of the micrometer-scale samples, resulting in the lack of spatial resolution and the disadvantage of detecting the magnetization of inhomogeneous sample synthesized at high pressure and high temperature inside the high-pressure chamber[10,11]. Integration of inductively coupled coils in high pressure chamber enables the measurement of macroscopic magnetism by the AC susceptibility method[12], however, it is still challenging to achieve sufficient sensitivity at megabar pressures.

In recent years, NV centers in diamond have been proposed as quantum sensors for new types of high-pressure magnetic measurements, as they can be placed at distances down to the nanoscales from the sample inside the pressure chamber to detect the complex free-space magnetic field textures generated by high-pressure samples and they are able to provide significant potential advantages in terms of sensitivity and spatial resolution[13–15]. However, the surrounding stress environment could dramatically degrade NV centers' magnetic sensing capabilities associated with their optical and spin properties. Recent works show that in complex stress environment, the optical-spin readout contrast worsens, and spin resonance linewidth becomes broader, leading to a sharp drop in sensitivity as the pressure approaches the megabar level[16,17]. Understanding the stress-induced effect and maintaining excellent performances of NV centers present substantial challenges for applying this technique to reveal the magnetically related physics above megabar.

In this work, we explore the impact of stress on the optical and spin properties of NV centers. Then by modulating the uniaxial stress along the nitrogen-vacancy axis, we improve the magnetic detective sensitivity to $\sim 1\mu T/\sqrt{Hz}$ at 130 GPa, which is promising for nano-scale single-domain grain detection[18,19]. Based on the sensitivity of the NV quantum sensors achieved, we investigated the magnetism change of the $Fe_3O_4$—one of the oldest magnetic minerals on the earth[20]—at pressures well exceeding a megabar at room temperature.

## Results and discussion

### Quantum sensors at megabar pressures

Firstly, to ensure the sensors work well at extreme pressure, an understanding of how the pressure alters the states of quantum sensors is necessary. In particular, the non-hydrostatic pressure components like shear stress have been found to greatly affect phase transition occurrence[21,22]. We investigated the electronic energy level of the NV center under uniaxial stress with different directions by first-principles calculations. As shown in Fig. 1b, the stress component $\sigma_\perp$ perpendicular to the NV axis breaks the $C_{3V}$ symmetry and lifts the degenerate energy levels $e_x$ and $e_y$. As the stress increases, the electrons will undergo redistribution, favoring staying in the same $e_x$ levels, resulting in a single state ($S = 0$) and loss of magnetic detection ability. In contrast, the symmetry is not broken for the stress applied along the NV axis, and the quantum sensor stays in the spin-triplet state ($S = 1$), enabling high-pressure magnetic measurements. This stress orientation-dependent spin crossover effect will provide insight into understanding spin-crossover phenomena in other materials at high pressures.

In experiments, the stress at the anvil tip comprises hydrostatic pressure ($\rho$) and uniaxial stress ($\sigma$) perpendicular to the surface[23],

allowing for modulation of uniaxial stress direction on NV centers through control of anvil crystal orientation. For NV centers oriented along [111] direction in the (111)-cut anvil, the component $\sigma_\perp$ can be minimized, in contrast to the increased component $\sigma_\perp$ for NV centers in the other three crystallographically equivalent directions ($[1\bar{1}\bar{1}], [\bar{1}1\bar{1}], [\bar{1}\bar{1}1]$). As a result, the contrast in the ODMR spectrum for [111] NV centers remains high as pressure increases to the megabar range (Fig. 2a), and the ODMR signal of NV centers in the other three directions disappear and loss sensing capability (Supplementary Fig. 6). The reason for the asymmetry in the ODMR peaks intensity is the asymmetry in effective driving strength due to $\sigma_\perp$ (see Supplementary Note 1, section 3.5). Notably, at 130 GPa, we observed a contrast enhancement to ~30%, much higher than that at 1.5 GPa (Fig. 2f). We suspect that NV centers oriented in the other directions, which produce intense optical background signals at low pressure, fluorescence quenching may occur at high pressure and thus increase the ODMR contrast (see Supplementary Note 2). In conclusion, the observed high contrast is approximately two orders of magnitude higher than previous results at similar pressures[16,17].

Secondly, we investigate the effect of inhomogeneous stress on the broadening of the ODMR spectrum ($\Gamma$). Together with the ODMR spectrum contrast ($C$), these two parameters are crucial in determining the sensitivity of magnetic measurements[24,25]:

$$\eta_B = \frac{4}{3\sqrt{3}} \frac{h}{g_e \mu_B} \frac{\Gamma}{C\sqrt{R}} \tag{1}$$

with the Planck constant $h$, g-factor $g_e$, Bohr magneton $\mu_B$, and photon-detection rate $R$.

As shown in Fig. 1c, d and Fig. 2b, in the presence of axial stress, the center frequency of $m_s = \pm 1$ spin sublevel shifts from $D_0$ (2.87 GHz) to $D_0 + D_s$[26,27]. In addition, inhomogeneous axial stress along the NV axis also broadens the ODMR spectrum and reduces the magnetic measurement sensitivity (see Supplementary Note 3). However, since the stress gradient is mainly perpendicular to the anvil surface and the distribution of the NV centers in the z direction is highly concentrated (~4 nm thickness through low-energy (9 keV) ion implantation), we can largely suppress the linewidth broadening. As shown in Fig. 2d, e, the linewidth of the ODMR spectrum is narrowed to approximately 20 MHz, a five-fold reduction compared to previous work[16,17]. With an external magnetic field projected along NV axis ($B_z$) and stress component $\sigma_\perp$, the energy splitting between $m_s = \pm 1$ sublevels can be expressed as $2\Delta = 2\sqrt{\Delta_B^2 + \Delta_{\sigma_\perp}^2}$, determined by the Zeeman splitting ($2\Delta_B = 2\gamma_e B_z$) and the stress splitting ($2\Delta_{\sigma_\perp}$). In this work, we minimize the $\sigma_\perp$ component using a (111)-cut anvil and obtain $\Delta \cong \Delta_B$ since $\Delta_B \gg \Delta_{\sigma_\perp}$ (Fig. 2c).

As shown in Fig. 2g, the improvement of ODMR contrast and the suppression of spectrum broadening result in a magnetic detection sensitivity of $\sim 1\mu T/\sqrt{Hz}$ at megabar pressure. Moreover, the sensor has a sub-microscale spatial resolution, as estimated from the optical diffraction limit of the confocal optical system. With such detection sensitivity and resolution, our quantum sensors can investigate the remnant magnetic field generated by nanoscale single-domain grains[18,19] at megabar pressures.

### Magnetic properties investigation at megabar pressures

As an illustrative example, the NV center quantum sensors are utilized to investigate the evolution of the magnetic properties of $Fe_3O_4$ across a wide pressure range, from ambient to megabar scale. As a widely studied and representative magnetic material in geomagnetism, its macroscopic magnetic behavior under high pressure remains not fully understood.

Previous studies have utilized various microscale detection techniques, such as neutron scattering, Mössbauer spectroscopy, Fe

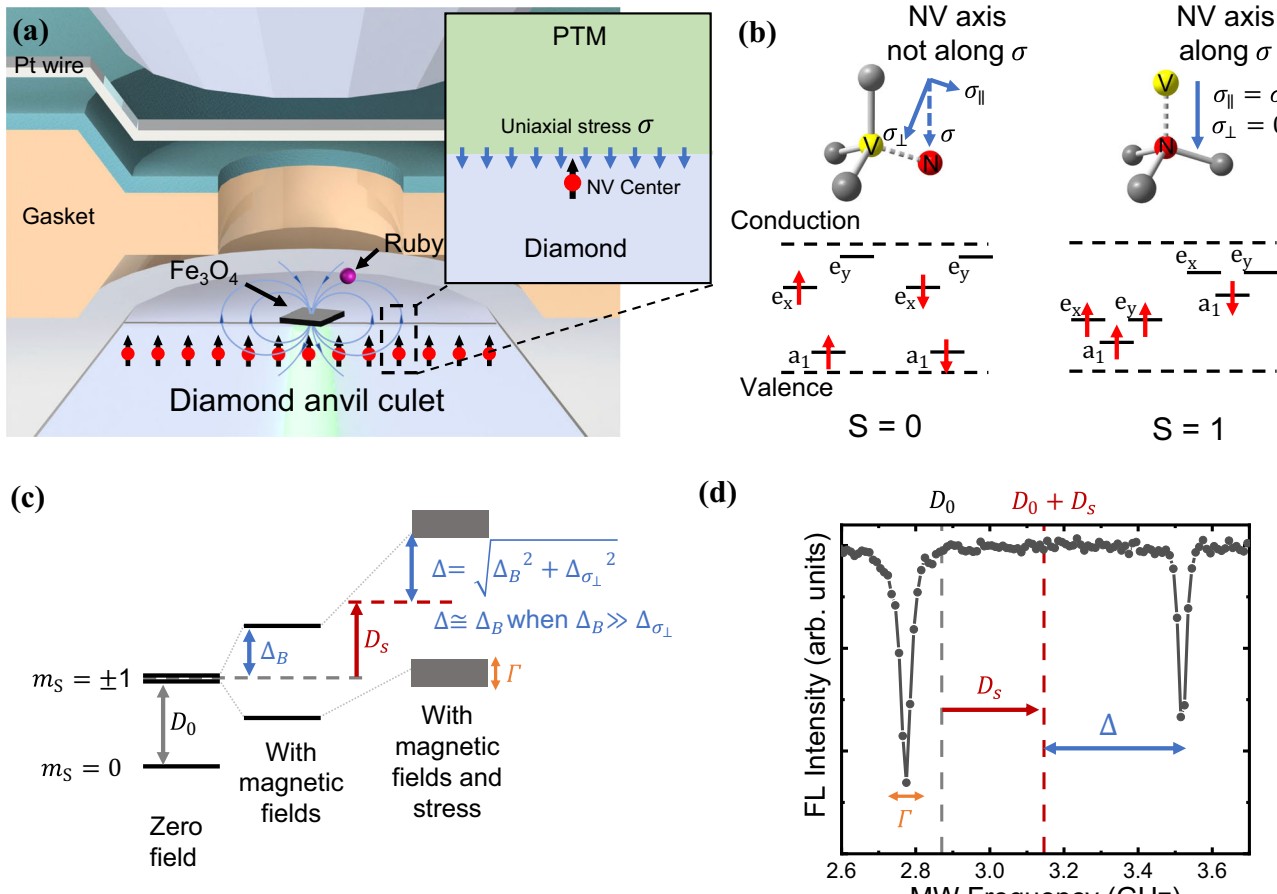

**Fig. 1 | Schematic geometry NV centers in the DAC. a** Schematic of the DAC geometry. The shallow layer of NV centers is embedded in the diamond anvil culet and a single crystal of $Fe_3O_4$ is placed on the surface of the same culet. The inset shows the uniaxial stress component $\sigma$ perpendicular to the surface of culet. **b** The electronic ground state levels of the NV center under megabar uniaxial stress along [111] direction by first-principles calculations. Left: NV centers along [$1\bar{1}\bar{1}$], [$\bar{1}1\bar{1}$],and [$\bar{1}\bar{1}1$] direction. Right: NV centers along [111] direction. **c** The spin ground state energy structure of the $NV^-$ and the evolution with the existence of both magnetic field and stress environment. The fluorescence intensity of the NV center is dependent on the spin states ($m_s = 0$, bright; $m_s = \pm 1$, dark). And the spin states can be initialized to $m_s = 0$ by laser and coherently manipulated by microwave. $D_0 = 2.87$ GHz is the zero-field splitting. $\Delta_B$ is half of the Zeeman splitting energy with external magnetic field. $D_s$ is the energy shift of the zero-field splitting caused by the axial stress component. $\Delta_{\sigma_\perp}$ is half of the splitting in $m_s = \pm 1$ spin sublevels produced by the $\sigma_\perp$ in NV's frame. $\Gamma$ is the additional broadening of the sublevels for the ensemble NV centers in inhomogeneous stress. **d** The corresponding optically detected magnetic resonance (ODMR) spectrum at 46.3 GPa presenting the spin states in (**c**). The dashed grey line marks the peak position ($D_0$) at ambient conditions with no magnetic field, and the dashed red line marks the central position ($D_0 + D_s$) of two separated peaks at high pressure with a parallel magnetic field (-133 G). The data extraction of $D_0$, $D_s$, $\Delta$, and $\Gamma$ allow us to determine the magnitude of the stress and external magnetic fields.

K-edge X-ray absorption, and magnetic circular dichroism measurements to infer the macroscopic magnetic behavior at high pressure[28–32]. These investigations focused on exploring spin crossover and exchange interactions at different atomic Fe sites. However, understanding the complex process of macroscopic magnetism involves the interplay of structural, spin transition, and magnetic orders, among other mechanisms. Obtaining a comprehensive understanding of macroscopic magnetism from a microscopic perspective is challenging and can lead to controversial conclusions[4,29–32]. In our study, according to direct stray field measurement, we observed the magnetic transition from ferrimagnetic to weak ferromagnetic and paramagnetic with the structural evolution from $\alpha$-$Fe_3O_4$ ($Fd\bar{3}m$) to $\beta$-$Fe_3O_4$ ($Bbmm$) and $\gamma$-$Fe_3O_4$ ($Pbcm$)[33]. These results address the controversy of magnetism in $\beta$-$Fe_3O_4$ ($Bbmm$) below 40 GPa[29,32] and exclude the possibility of magnetism recovery from 65 GPa to 120 GPa at room temperature[4].

The pressure dependence of magnetic properties is characterized by the ODMR spectra of two kinds of NV centers (Fig. 3a, b). The $NV_o$, located approximately 1 μm away from the $Fe_3O_4$, acts as a high-dynamical pressure range magnetometer for detecting magnetite-induced static stray fields. These stray fields cause additional spectral shifts relative to the external constant bias magnetic field (Fig. 3c). On the other hand, $NV_i$ positioned a few nanometers beneath the $Fe_3O_4$, serves as a high-precision and high spatial resolution magnetometer. The linewidth broadening of the ODMR spectrum in $NV_i$ is attributed not only to inhomogeneous stresses but also to strong spatial magnetic field gradients and potential magnetic fluctuations[34,35] near the surface of magnetite (Fig. 3d).

As illustrated in the schematic figure (Fig. 4a), variations in magnetization strength and configuration under pressure lead to significant changes in the stray magnetic field, which can be detected by NV centers. The magnetic evolution of magnetite from ferrimagnetic (FiM) to weak ferromagnetic (FM) and finally to paramagnetic (PM) is depicted in the high-pressure phase diagram (Fig. 4e) based on measurements of the stray magnetic fields (Fig. 4b–d). Magnetic imaging further enriches the details of the orientation and distribution evolution of magnetic domains (Fig. 4f, g).

In $\alpha$-$Fe_3O_4$, we observe the orientation of magnetic domains undergo dynamic changes as pressure increases. This results in a distinct evolution of the stray magnetic fields associated with different

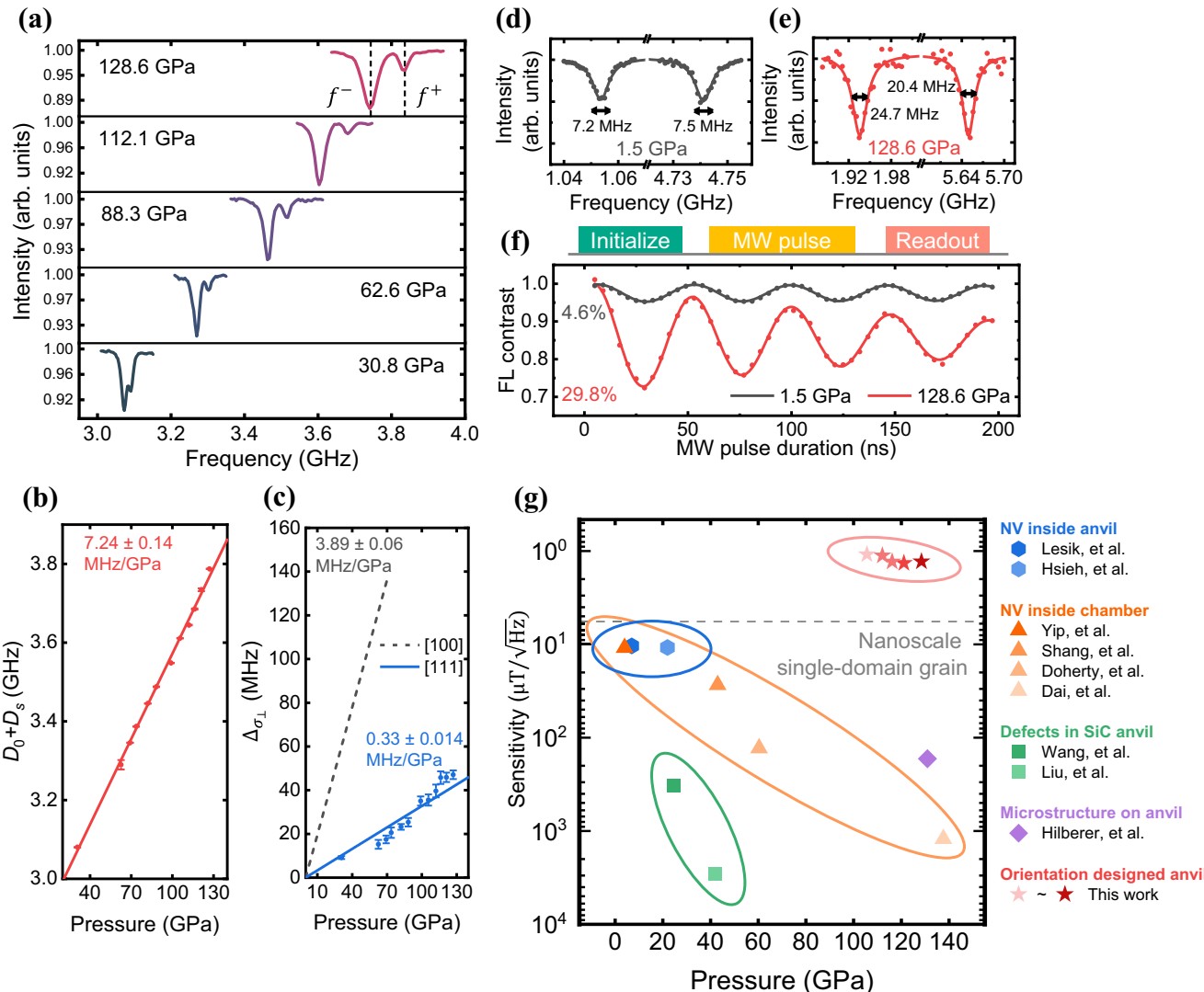

**Fig. 2 | NV centers in crystal orientation designed diamond anvil. a** Zero magnetic field ODMR spectrum of NV centers under pressure. $f^{\pm}$ are the resonant frequencies of the left and right peaks in the spectral line respectively, ODMR center frequency is $D_0 + D_s = (f^+ + f^-)/2$, and ODMR splitting induced by $\sigma_\perp$ is $2\Delta_{\sigma_\perp} = f^+ - f^-$. Please note that due to the ODMR not being performed under optimized conditions, there is a reduction in contrast compared to the results shown in (**f**). **b** Pressure dependence of $D_0 + D_s$. The data are represented by red filled circles with error bars, and the red solid line represents the linear fitting of the data. **c** Pressure dependence of $\Delta_{\sigma_\perp}$, which is measured by the half of splitting in ODMR peaks in zero magnetic field ($\Delta_B = 0$). The blue filled circles with error bars represent the data in (111)-cut anvil, and the blue solid line provides a trendline for the deviation of $\Delta_{\sigma_\perp}$ from linearity. The dashed grey line represents the result in (100)-cut anvil by Hilberer et al[16]. The error bars in (**b**) and (**c**) correspond to a 95% confidence interval. **d**, **e** The ODMR spectrum linewidth ($\Gamma$) tested in ~660 G magnetic field at 1.5 GPa and 128.6 Gpa. The spectrum broadening is shown respectively. The $Y$ axes are set equal scales for comparison. **f** Rabi oscillations of NV electron spins at 1.5 GPa and 128.6 Gpa. The ODMR contrast is shown respectively. The experimental data are fitted by damped sine wave function and plotted by the gray and red solid lines. **g** Comparison of high-pressure magnetometry techniques based on color centers in diamond anvil cell. Our work is marked by red stars. The performance of NV inside the diamond anvil (blue hexagons)[13,14], NV inside the pressure chamber (orange triangles)[15,17,46,47], microstructure on anvil (violet diamonds)[16] and defects in SiC anvil (green squares)[48,49] are plotted together for comparison. The dashed grey line indicates the magnitude of the remnant magnetic field (~5.8μT) generated by a nanoscale single-domain grain (magnetic moment $10^{-17}$A·m², distance 700nm).

magnetic domains, which can be observed using NV$_o$ at different sites (Fig. 4b). It is important to note that this evolution is accompanied by an overall reduction in the magnetic strength observed at all sites, especially as the first structure transition occurs around 30 GPa. This reduction in magnetic strength is confirmed by two independent measurements of the stray fields from two different samples of single crystal magnetite (Fig. 4c).

In $\beta$-Fe$_3$O$_4$, the macroscopic magnetism experiences a significant decrease, resulting in stray magnetic fields that are too weak to be detected using NV$_o$ (Fig. 4b, c). In contrast, the NV$_i$ beneath the Fe$_3$O$_4$ demonstrates the advantage in measuring and imaging such weak stray magnetic fields (Fig. 4f–j). The presence of both stray

magnetic fields and the linewidth broadening (Fig. 4d) indicate that $\beta$-Fe$_3$O$_4$ exhibits weak ferromagnetism. The drastic reduction of magnetism in comparison to $\alpha$-Fe$_3$O$_4$ also explains the disappearance of ODMR spectra in NV$_i$ at 0-30 GPa (Fig. 3d) due to excessive line broadening caused by strong spatial magnetic field gradient and potential magnetic fluctuations. Additionally, the images reveal that the magnetic domains tend to be opposite to each other. The weak ferromagnetism observed in $\beta$-Fe$_3$O$_4$ can be influenced by various complex factors such as altered magnetic interactions and spin crossover. Ab initio calculations further support the possibility of the presence of multiple complex magnetic orders and spin transitions in $\beta$-Fe$_3$O$_4$[36].

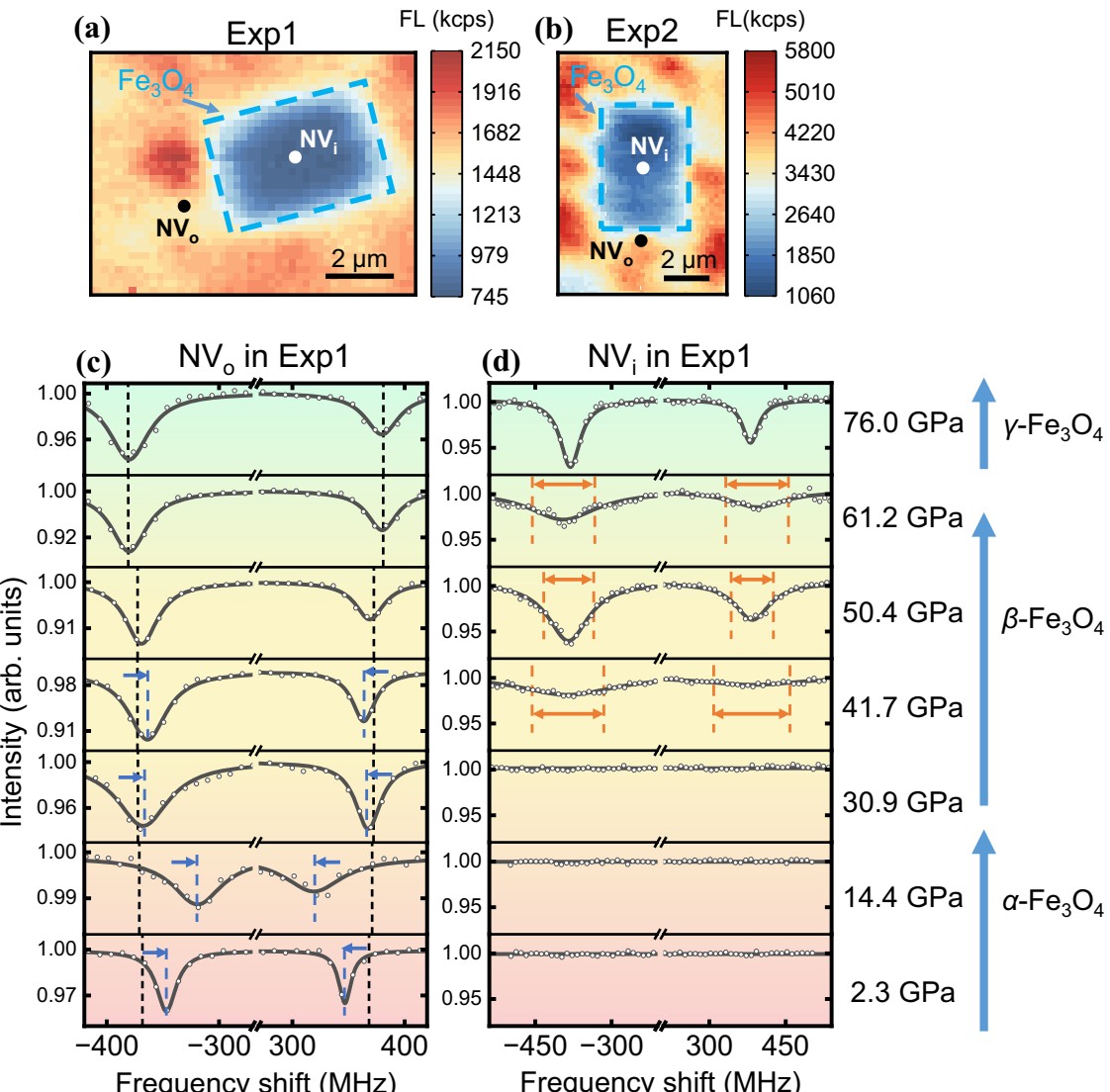

**Fig. 3 | Data for magnetic detection of magnetite. a, b** The Fluorescence (FL) imaging of NV centers in DACs exp1 and exp2, respectively. The fluorescence intensity tends to be lower beneath the $Fe_3O_4$ sample. The dashed blue square represents the area of $Fe_3O_4$ sample for eye guide. The filled black and white circle marks the NV centers below the side ($NV_o$) and beneath ($NV_i$) the $Fe_3O_4$ sample, respectively. **c, d** The ODMR spectra of $NV_o$ and $NV_i$ change with elevated pressures, respectively. The horizontal axis represents the frequency shift relevant to the center frequency of $m_s = \pm 1$ spin sublevels. The dashed black lines mark the resonant frequency of the NV centers far away from the sample, which is used to characterize the magnitude of the external magnetic field along NV axis (-130 G). The dashed blue lines and arrows mark the resonant frequency shift induced by the magnetic field from the sample. The dashed orange lines and arrows mark the spectrum broadening.

In $\gamma$-$Fe_3O_4$, we observe the disappearance of both the stray magnetic fields and the linewidth broadening above 70 GPa (Fig. 4d), indicating a magnetic transition to PM phase. This observation is consistent with the Mössbauer spectroscopic studies that suggest magnetite becomes paramagnetic above 70 GPa at room temperature[28,37]. Interestingly, before this magnetic transition occurs, we observe the fragmentation in magnetic domains around 60 GPa (Fig. 4h–j) resulting in an increase of the line-broadening in ODMR spectra (Fig. 4d). We extend the magnetism measurements beyond 120 GPa and no further magnetic transition is observed, ruling out the possibility that the magnetism might recover above 65 GPa at room temperature[4].

Our work extends magnetic measurement techniques based on NV centers to the megabar pressure range. It provides a new tool for quantitatively analyzing macroscopic magnetism at extreme pressures. A combination of complex factors such as electron spins, magnetic interactions, and crystal structures at the microscopic

molecular scale determines the evolution of the magnetic properties of materials under extreme pressure. Utilizing the improved performance of the quantum sensors, we have realized the direct observation of magnetite's magnetism under extreme pressure in various complex situations. The high spatial resolution and high sensitivity magnetic measurement can overcome the inhomogeneity of the samples and reveal the local magnetic evolution on the micrometer and nanometer scales.

This technique also applies to in-situ direct detection of high-temperature superconducting materials under megabar pressure, including various super hydride compounds[7,38]. We note a contemporaneous work using NV centers on the (111)-cut anvil for the detection of the Meissner effect in the $CeH_9$ hydride superconductor[39]. In addition to the DC magnetic field measurements, our NV centers' optical and spin properties also lift the limitations of nuclear magnetic resonance (NMR) and electron spin resonance (ESR) measurements under megabar pressures[40]. Thus, this work is promising to enable

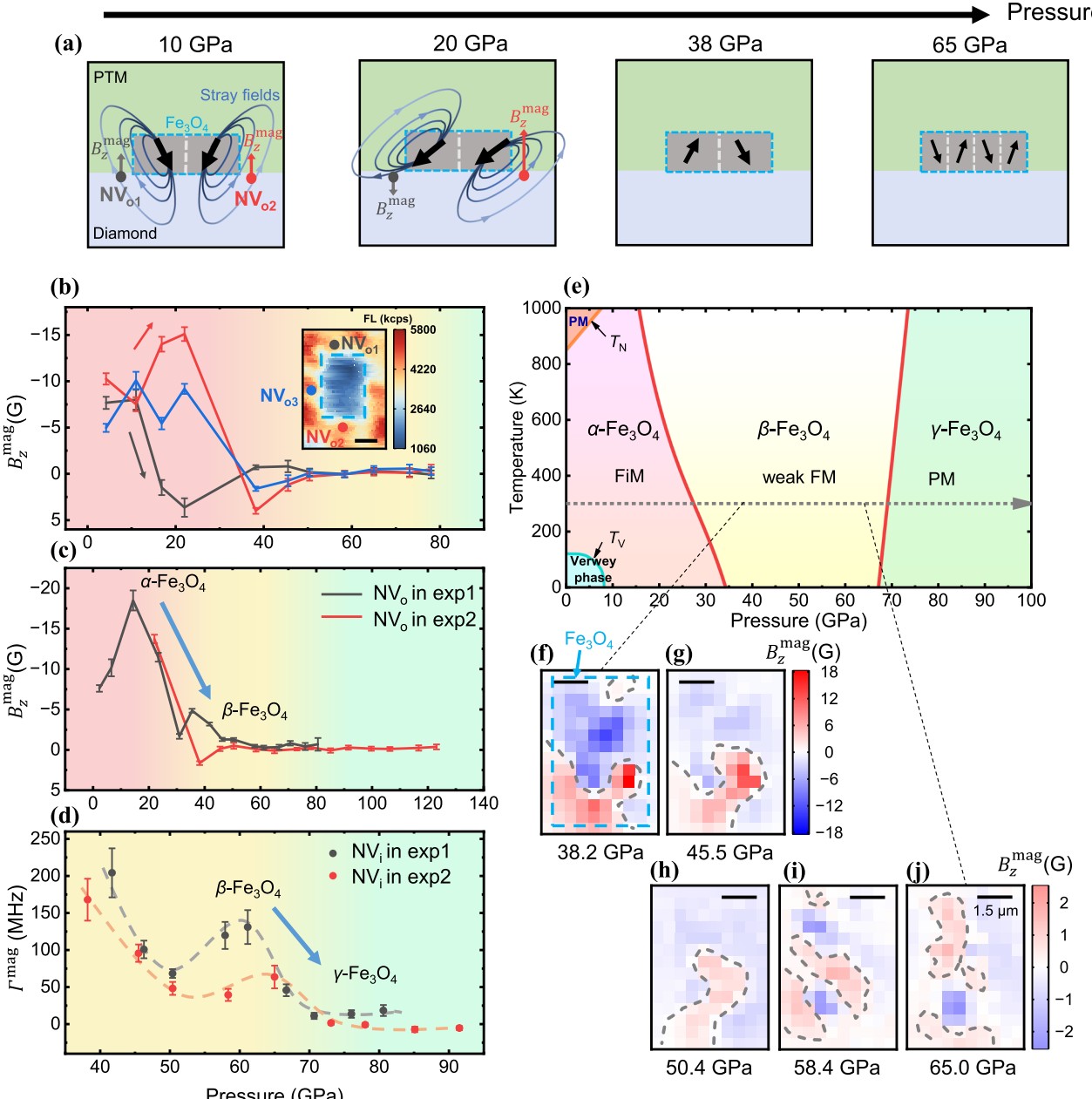

**Fig. 4 | Magnetism evolution of magnetite to megabar pressures. a** Schematic diagram of the evolution of magnetic domains and their stray magnetic fields in magnetite with pressure. **b, c** Pressure dependence of the magnetite magnetic field $B_z^{mag}$. In (**b**), the black, red, and blue lines indicate the magnetic field $B_z^{mag}$ with an external magnetic field of -240 G at positions $NV_{o1}$, $NV_{o2}$, and $NV_{o3}$ in exp2, respectively, in the inset. The dashed blue line in the inset marks the $Fe_3O_4$ sample. **c** Black and red lines show the magnetic field $B_z^{mag}$ in exp1 (at positions $NV_o$ in Fig. 3a) and exp2 (at positions $NV_o$ in Fig. 3b) with an external magnetic field of -130 G and -570 G along NV axis respectively. The external magnetic fields used in exp1 and exp2 are to ensure the $\Delta_B \gg \Delta_{\sigma_\perp}$ condition in 80 GPa and 120 GPa, respectively. (See Supplementary Note 4, section 6.4 for details of measurement deviation) (**d**) Pressure dependence of the linewidth broadening $\Gamma^{mag}$. The black and red dots show the data from exp1 (at positions $NV_i$ in Fig. 3a) and exp2 (at positions $NV_i$ in Fig. 3b), respectively. The dotted lines are guide lines describing the trend of changes. The error bars in (**b**), (**c**), and (**d**) correspond to a 95% confidence interval. **e** The phase diagram of $Fe_3O_4$[30]. FM stands for ferromagnetic, FiM stands for ferrimagnetic, and PM stands for paramagnetic. **f–j** Magnetic field imaging of the surface of $Fe_3O_4$ in exp2 with an external magnetic field of -240 G at pressures of 38.2, 45.5, 50.4, 58.4 and 65.0 GPa, respectively. The dashed blue line in (**e**) marks the $Fe_3O_4$ sample and dashed gray lines mark the magnetic domain wall. (The details of error analysis are in Supplementary Note 4).

magnetic resonance imaging (MRI) in pressure chambers, thus providing an opportunity to make a great impact in investigating high-temperature superconductivity at high pressures[6,7,41–43].

## Methods

In our experiments, we used a BeCu symmetric diamond anvil cell (Fig. 1a) compressing rhenium gasket to provide a high-pressure environment for $Fe_3O_4$ samples. 532 nm laser is used to optically initialize and read out the spin state of NV centers. Ruby fluorescence and diamond edge Raman are used to calibrate pressures in DAC chamber. A platinum wire was compressed between the gasket and anvil pavilion facets and served as the microwave radiation guide for controlling the spin state. Nano cBN mixed with epoxy powder isolated the Pt foil and gasket. The sample chamber confined by the gasket and diamond culets was filled with pressure-transmitting medium (PTM) KCl to provide a quasi-hydrostatic environment. The magnetite sample

($Fe_3O_4$ single crystal) was placed on the diamond surface and compressed together with PTM. Ruby fluorescence (below 20 GPa) and Raman spectra of the diamond (above 20 GPa) were used as pressure calibrants[44,45].

The diamond anvils we used in experiments were cut and polished from HPHT type-IIas (Non-fluorescent) single crystal diamonds. Both 100 μm and 150 μm diameter the culets were used. The NV centers layer used for magnetic sensing was positioned about ~9 nm deep from the surface of the anvil. The NV centers layer was created by $^{14}N^+$ ion implantation (Energy: 6 keV Dose: $1 \times 10^{13}/cm^2$) and vacuum annealing treatment (1000 °C -2 h). The size of the magnetite sample was about 4 μm × 5 μm × 1 μm, which was cut from a bulk 99.99% purity single crystal $Fe_3O_4$ (HeFei Crystal&Surface Technical Material Co., Ltd).

## Data availability
All data and code that support the findings of this study are available from the corresponding author upon request.

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

## Acknowledgements

The fabrication of diamond sensors was partially performed at the USTC Center for Micro and Nanoscale Research and Fabrication, and the authors particularly thank Dr. Xiaolei Wen, Xiwen Wang, Cunliang Xin, and Hongfang Zuo for their assistance in Fe$_3$O$_4$ and sensors fabrication process. Alexander F. Goncharov thanks for the support of Carnegie Science. This work is supported by the National Natural Science Foundation of China (Grants No.T2325023, 92265204, 12104447, 12204485, 62304216), the CAS (GJJSTD20200001), the National Key R&D Program of China (Grants No.2021YFB3202800, 2023YFF0718400, 2023YFB3209901), the Innovation Program for Quantum Science and Technology (Grant No. 2021ZD0302200), the Anhui Initiative in Quantum Information Technologies (Grant No. AHY050000), the Postdoctoral Fellowship Programof CPSF (Grant No. GZB20240717), the China Postdoctoral Science Foundation (Grant No. 2024M753084), the Fundamental Research Funds for the Central Universities.

## Author contributions

J.D., Ya.W., and Yu.W. supervised the project, Ya.W., Yu.W., and M.W. proposed the idea of the experiment. M.W. and Z.L. built the experimental setup and performed the measurements. Yu.W. and G.X. prepared the diamond anvil cell integrated quantum sensor. M.W., P.Y., H.S., and X.Y. prepared the NV center and the magnetite sample. B.Y. performed a first-principles calculation. M.W., Yu.W., Z.L., G.X., and Ya.W. performed the data analysis and wrote initial draft of the manuscript. A.F.G. and J.Z. reviewed and revised the manuscript. All authors discussed the results and commented on the manuscript.

## Competing interests

The authors declare no competing interests.
