## [Peer Review File · Nature Communications]

Imaging magnetic transition of magnetite to megabar pressures using quantum sensors in diamond anvil cellEditorial Note: This manuscript has been previously reviewed at another journal that is not operating a transparent peer review scheme. This document only contains reviewer comments and rebuttal letters for versions considered at *Nature Communications*.

REVIEWERS' COMMENTS

Reviewer #1 (Remarks to the Author):

In this work, the authors demonstrated significantly improved ODMR contrast and sensitivity of quantum sensing under high pressure in a diamond anvil cell by utilizing [111]-cut diamond. Additionally, the authors employed this technique to image the magnetic field of magnetite and investigate its magnetic transitions. We appreciate the authors' great efforts in addressing all the previous referees' comments. We believe the manuscript in its current form will be of great interest to the broad readership of Nature Communications. We are happy to recommend the manuscript for publication.

One small suggestion for the authors to consider: In Fig. 3c, the readability would be improved if the authors could adjust the range of y axes to be the same for different pressure points, so that one can directly visualize the contrast change (broader transition leads to lower contrast). The same applies to Fig 3d.

One typo: In Line 139 of the main text, (Fig. S3) should be (Fig. S6)

Reviewer #2 (Remarks to the Author):

The authors have addressed all the concerns raised by the reviewers and I am pleased to recommend this paper for publication in Nature Communications.

Two minor remarks are in order.

(1) In an ideal (111)-cut anvil and under ideal pressure loading, there will be no uniaxial stress. However, the experiment shows a splitting of 0.33 ± 0.01 MHz/GPa caused by uniaxial stress, which is detrimental to the zero-field application of NV-based quantum sensing. A discussion on the origin and method to further optimize the uniaxial stress would be beneficial.

(2) Line 290, “The drastic reduction of magnetism in comparison to α -Fe₃O₄ also explains the disappearance of ODMR spectra in NVi at 0-30 GPa ...” It will be helpful to cite Fig. 3d there, as this has not yet been discussed.

Reviewer #3 (Remarks to the Author):

-----**Reply to Reviewer #1**-----

Reviewer #1 (Remarks to the Author):

In this work, the authors demonstrated significantly improved ODMR contrast and sensitivity of quantum sensing under high pressure in a diamond anvil cell by utilizing [111]-cut diamond. Additionally, the authors employed this technique to image the magnetic field of magnetite and investigate its magnetic transitions. We appreciate the authors' great efforts in addressing all the previous referees' comments. We believe the manuscript in its current form will be of great interest to the broad readership of Nature Communications. We are happy to recommend the manuscript for publication.

One small suggestion for the authors to consider: In Fig. 3c, the readability would be improved if the authors could adjust the range of y axes to be the same for different pressure points, so that one can directly visualize the contrast change (broader transition leads to lower contrast). The same applies to Fig 3d.

Reply:

We sincerely appreciate the valuable feedback from the reviewer. In response to their helpful suggestions, we have adjusted the y-axis range for the data points in Fig. 3d to be the same. This modification makes it more visual to show that the broader transition leads to lower contrast, ultimately enhancing readability. The same adjustment is not applied to Fig. 3c, as it might obscure the CW peak positions at some pressure points, hindering readers from accessing critical information.

One typo: In Line 139 of the main text, (Fig. S3) should be (Fig. S6)

Reply:

We thank the reviewer for pointing out the typo. We have corrected it in the main text.

-----**Reply to Reviewer #2**-----

Reviewer #2 (Remarks to the Author):

The authors have addressed all the concerns raised by the reviewers and I am pleased to recommend this paper for publication in Nature Communications.

Two minor remarks are in order.

(1) In an ideal (111)-cut anvil and under ideal pressure loading, there will be no uniaxial stress. However, the experiment shows a splitting of 0.33 ± 0.01 MHz/GPa caused by uniaxial stress, which is detrimental to the zero-filed application of NV-based quantum sensing. A discussion on the origin and method to further optimize the uniaxial stress would be beneficial.

Reply:

We sincerely appreciate the feedback from the reviewer! In response to it, we have added the following discussion in SI section 6.4 the last paragraph:

“Considering the adverse effects of the σ_{\perp} component on the zero-filed application of NV-based quantum sensing, it may be possible in the future to optimize the uniaxial pressure by employing an ideal (111)-cut anvil or using gas pressure-transmitting media such as Ne or Ar. This optimization may further suppress the generation of $\Delta\sigma_{\perp}$.”

(2) Line 290, “The drastic reduction of magnetism in comparison to α -Fe₃O₄ also explains the disappearance of ODMR spectra in NVi at 0-30 GPa ...” It will be helpful to cite Fig. 3d there, as this has not yet been discussed.

Reply:

We thank the reviewer for the helpful suggestion. We have cited Fig. 3d in the sixth paragraph of section “Magnetic properties investigation at megabar pressures” in main text.

-----**Reply to Reviewer #3**-----

Reviewer #3 (Remarks to the Author):

Reply:

We appreciate the comprehensive review provided by the reviewer.

----- **List of changes** -----

1. We add an affiliation: Anhui Province Key Laboratory of Scientific Instrument Development and Application, ranked third in the list.
2. Following your checklist guidance, we removed the Conclusion heading and merged the last two paragraphs into Results and Discussion section;
3. We changed the format of some variable symbols and mathematical terms in the manuscript and Supplementary Information to conform the checklist guidelines. The symbols in the Figures is amended accordingly.
4. In the manuscript and Supplementary Information, equations are labeled sequentially as (1), (2), (3), etc.
5. We have shortened the figure captions in the manuscript, to stay within the word count requirement.
6. We follow reviewer #1's suggestion and adjust the y-axis range for the data points in Fig. 3d to be the same
7. We follow reviewer #2's suggestion 1 and add a discussion on the origin and method to further optimize the uniaxial stress in the last paragraph of Section 6.4 in the Supplementary Information
8. We follow reviewer #2's suggestion 2 and cited Fig. 3d in the sixth paragraph of section “Magnetic properties investigation at megabar pressures” in main text
9. We have changed all the (a. u.) into (arb. units).
10. In Data Availability Statement, we changed "upon reasonable request" into "upon request".
11. We have changed the “Data and Code Availability” into “Data availability”.